# Management of Amelogenesis Imperfecta in Childhood: Two Case Reports

**DOI:** 10.3390/ijerph18137204

**Published:** 2021-07-05

**Authors:** Mirja Möhn, Julia Camilla Bulski, Norbert Krämer, Alexander Rahman, Nelly Schulz-Weidner

**Affiliations:** 1Dental Clinic, Department of Pediatric Dentistry, Justus Liebig University, Schlangenzahl 14, 35392 Giessen, Germany; julia.c.bulski@dentist.med.uni-giessen.de (J.C.B.); norbert.kraemer@dentist.med.uni-giessen.de (N.K.); nelly.schulz-weidner@dentist.med.uni-giessen.de (N.S.-W.); 2Department of Conservative Dentistry, Periodontology and Preventive Dentistry, Hannover Medical School, Carl-Neuberg-Strasse 1, 30625 Hannover, Germany; rahman.alexander@mh-hannover.de

**Keywords:** amelogenesis imperfecta, pediatric dentistry, dental care, therapy concept

## Abstract

Amelogenesis imperfecta (AI) is defined as an interruption of enamel formation due to genetic inheritance. To prevent malfunction of the masticatory system and an unaesthetic appearance, various treatment options are described. While restoration with a compomer in the anterior region and stainless steel crowns in the posterior region is recommended for deciduous dentition, the challenges when treating such structural defects in mixed or permanent dentition are changing teeth and growing jaw, allowing only temporary restoration. The purpose of this case report is to demonstrate oral rehabilitation from mixed to permanent dentition. The dentition of a 7-year-old patient with AI type I and a 12-year-old patient with AI type II was restored under general anesthesia to improve their poor aesthetics and increase vertical dimension, which are related to problems with self-confidence and reduced oral health quality of life. These two cases show the complexity of dental care for structural anomalies of genetic origin and the challenges in rehabilitating the different phases of dentition.

## 1. Introduction

Amelogenesis imperfecta (AI) is described as generalized defects in enamel formation in primary and permanent dentition because of a genetic disorder. The inherited malformation of teeth can be x-linked, autosomal dominant, autosomal recessive, or sporadic. In particular, mutation or altered expression of the enamelin (ENAM), amelogenin (AMEL), matrixmetalloproteinaise-20 (MMP20), kallikrein-4 (KLK4), and FAM83H genes is associated with the malfunction of enamel-forming proteins [1]. An association with a general or systematic disorder has not been reported.

The clinical manifestation includes four types of AI [2]. The most common phenotype is type I, characterized by a hypoplastic structure with a decreased quantity of enamel. The teeth show reduced enamel thickness, rough surface, and various extensions of defects (Figure 1) [3]. Type II, called hypomaturation, shows mottled and softer enamel due to defective protein maturation within the enamel matrix. Additionally, chipping of the enamel from the dentin can be found (Figure 2) [1]. In AI type II, the enamel thickness is normal. The secretion phase of the ameloblasts proceeds as usual, but in the maturation phase, normal reabsorption of the secreted enamel matrix proteins does not take place. Subsequently, a very high proportion of organic matter remains in the enamel [4].

Type III (hypocalcification) is associated with defects in calcification and appears in enamel with normal thickness at the time of eruption. Because of the poor mineralization, the enamel rapidly wears down and X-rays show less opacity. Type IV manifests as a mixed appearance of hypoplasticity–hypomaturation combined with taurodontism [1,5].

Due to the structure of enamel hypersensitivity, plaque accumulation and poor aesthetics are reported [6]. To prevent dental caries, gingival inflammation, open bite, or loss of vertical dimension, interdisciplinary patient care is recommended. In particular, conservative or prosthetic and orthodontic treatment are crucial for successful oral rehabilitation. Various treatment options have been described depending on the patient’s age and socioeconomic conditions and the severity of malformation [7]. While stainless steel crowns, strip crowns, and compomer restorations are common in primary dentition, the challenge in mixed and permanent dentition in adolescents is care of the dentition during growth [8]. Whereas ceramic crowns and veneers are preferred for adults, CAD/CAM composites offer an opportunity for high-quality restorations in adolescent children. The advantages of this approach are less chair time and the possibility of intraoral repairs in cases of material fractures. Besides the oral complications due to the genetic defects of the enamel, poor aesthetics can also be associated with problems with self-confidence and reduced oral health-related quality of life [9,10].

The aim of this paper was to report the management of AI patients from mixed dentition in childhood to permanent dentition in early adulthood by presenting two patients, aged 7 and 12 years. For this purpose, the differences between direct filling therapies combined with prefabricated crowns in mixed dentition and indirect restoration in the permanent dentition were compared, and the patients were followed up at 3 and 6 months.

## 2. Case Reports

### 2.1. Case Report 1: Mixed Dentition

A 7-year-old girl was referred to our pediatric polyclinic due to aesthetic problems and sensitive teeth with pain. The mother felt extremely affected in her social life by her daughter’s structural problem, as the child was being teased by other children at school due to her teeth. The family history showed no abnormalities; neither parent had phenotypic dental structure anomalies. Clinical examination revealed AI type I hypomineralized teeth. Oral examination presented easily chipping enamel combined with reduced enamel thickness, rough surface, and various extensions of structural loss. Defects in enamel matrix formation showed pitted and grooved enamel, especially in the maxillary front and upper and lower first permanent molars. In addition, the maxillary and mandibular fronts were clearly spaced apart. The patient’s oral hygiene was inadequate due to hypersensitivity and the tooth surface structure. A panoramic radiograph revealed loss of enamel (Figure 3). Tooth wear could be detected (Figure 4). A caries lesion in region 64 could be diagnosed (Figure 5a–d).

Due to the extensive scope of care, the patient underwent comprehensive rehabilitation, which was performed under general anesthesia because of her age and anxiety. Adhesive build-ups and stainless steel crowns stabilized the vertical dimension.

Stainless steel crowns (3M^™^, Neuss, Germany) were applied to second primary molars; after tangential preparation, adaptation, and control of the “snap effect”, metal crowns were cemented with glass ionomer cement (Ketac^™^ Cem Aplicap^™^, 3M^™^, Neuss, Germany). First primary molars were reconstructed using an all-in-one adhesive system (Scotchbond^™^ Universal, 3M^™^, Neuss, Germany) and compomer (Dyract^®^, Dentsply Sirona, Bensheim, Germany), except tooth 64, which was severely destroyed and had to be extracted. Permanent molars and anterior teeth were covered with direct composite filling material (Figure 5e–h). Without removing malformed enamel, etching with 34% phosphoric acid (Scotchbond^™^ Etchant, 3M^™^, Neuss, Germany) for 60 s was carried out, followed by rinsing and drying. After bonding (Scotchbond^™^ Universal, 3M^™^, Neuss, Germany), a flow composite (Venus^®^ Diamond Flow A2, Kulzer, Hanau, Germany) was directly applied, and then, composite (Venus^®^ Diamond A2, Kulzer, Hanau, Germany) was applied.

After rehabilitation, the patient was reevaluated after 3 months. Follow-up 6 months later showed a satisfactory aesthetic and functional result. Since there was no space narrowing in region 74, we did not need to fabricate a space maintainer (Figure 6).

### 2.2. Case Report: Permanent Dentition

A 12-year-old patient presented to our polyclinic because of the unattractive aesthetics of his teeth due to AI type II. In addition to the change in color of all permanent teeth, the loss of vertical dimension and the gaps in the dentition were impressive (Figure 7). The 12-year molars had not yet (completely) erupted into the oral cavity. In addition to AI, the patient presented with cardiac disease and immunodeficiency. As he was a foster child, his family history was not available.

Due to the strong need for aesthetic improvement and the associated general disease, it was decided to perform all-composite rehabilitation. In this direct approach, resin composite restoration was used for the transitional treatment of hypomature AI.

First, an impression of the maxilla and mandible with A-silicone material (Panasil^®^ Putty Fast, Kettenbach, Germany) was taken through advanced chairside. Due to the sufficient space of the upper and lower jaws, there was no need for preparation. Full composite crowns were built up on plaster models after articulation according to average values by the dental technician. A jaw relation determination could not be performed due to the patient’s insufficient compliance. Due to this particular feature, it should be pointed out, in this case, that adaptation to the new mandibular posture would be required after insertion. The parents were informed about this in detail.

Under general anesthesia, the restorations were tried on and cemented with dual-curing resin cement (Variolink^®^ Esthetic, Ivoclar^™^ Vivadent^™^, Ellwangen, Germany) using Monobond^®^ Plus and Adhese^®^ Universal VivaPen^®^ (Ivoclar^™^ Vivadent^™^, Ellwangen, Germany). The occlusion was slightly raised to provide sufficient space for this restorative reconstruction. Fissure sealing (Helioseal^®^, Ivoclar^™^ Vivadent^™^, Ellwangen, Germany) was performed on the second permanent molars, and stainless steel crowns (3M^™^, Neuss, Germany) were placed on all first permanent molars and cemented with glass ionomer cement (Ketac^™^ Cem Aplicap^™^, 3M^™^, Neuss, Germany) (Figure 8).

Three-month follow-ups were recommended to identify and repair possible defects at an early stage. In our case, the patient adapted to full crown treatment very well and had no symptoms in the temporomandibular joint after follow-up within 6 months.

## 3. Discussion

Rehabilitating a patient with AI is challenging from both the functional and aesthetic point of view. The complexity of the disease requires an interdisciplinary approach to achieve optimal treatment results. Several treatment options have been proposed. Recently, the use of bonded restorations has gained popularity due to the many advantages of these materials, including excellent aesthetics, conservative approach, and improved wear make.

Dental rehabilitation is one important part of improving oral health-related quality of life for children with generalized structure defects. The main objective of dental treatment of patients with hereditary structural anomalies is to prevent nearby caries damage [6]. In addition, the dentist should counteract the abrasion of the clinical crown by performing early treatment in order to prevent dimension loss and tooth loss [3]. In every case, the age of the patient must be considered in treatment planning. In our case, aesthetic rehabilitation was very important for both patients.

According to Toupenay et al., there is no agreement regarding the protocol for therapy except the timing: treatment should begin as early as possible to prevent tooth sensitivity and enamel loss [3].

While stainless steel crowns, strip crowns, and compomer restorations are common in primary dentition, the growing jaw and the changing of teeth present challenges in terms of treatment options in mixed and permanent dentition. Therefore, the full spectrum of dental materials should be exhausted during the development of a young adolescents. In particular, primary and mixed dentition only allow temporary therapy: conventional/resin-based glass ionomer cements, compomers/composites, strip crowns, preformed metal, or tooth-colored crowns [11].

In our first case, we mainly chose direct adhesive filling materials for tooth build-up because of the small loss of substance. It is typical for AI type I that the existing malformed enamel has similar or identical characteristics to enamel that is formed regularly [1]. For adhesive therapy, this means that a normal etching pattern can be expected and the adhesive system used will act identically to physiologically formed enamel. Therefore, it was not necessary to remove the enamel partially or completely in this case. Especially in younger patients with teeth that have just erupted, it can be beneficial for the practitioner and the patient to be able to restore them noninvasively but still functionally and aesthetically. Conventional or resin-based glass ionomer cements would not have been suitable due to their lower flexural strength and wear resistance. These properties allow the material to be used temporarily chairside, but it should not be applied under optimal conditions when using general anesthesia [12]. The development of compomers (polyacrylic/polycarboxylic acid modified composites) combines the advantages of glass ionomer cements (easy application) and composites (aesthetics). Nowadays, compomers (e.g., Dyract^®^) are the first choice for restoring primary teeth. In contrast, composites should be used as a long-term filling material in permanent dentition because of their higher wear resistance and compressive, flexural, and tensile strength [13]. Only the second primary molars were restored with stainless steel crowns to repair circular defects in this case. The advantages of metal crowns are easy adaptation, gentle preparation, and time-saving handling [14]. Preformed ceramic crowns for molars and incisors require strict preparation and involve high abrasion of antagonists [15]. In follow-up, we could not find any loss of filling, which was also to be expected regarding the etching pattern, which did not differ from the healthy enamel.

For affected permanent teeth, various full crowns are indicated. Depending on the patient’s age, metal, composite, and ceramic crowns are common. While individual ceramic crowns are contraindicated in adolescents due to jaw growth, composites can provide a temporary restoration. In particular, newly introduced high-performance CAD/CAM composites enable aesthetic restoration of malformed permanent teeth in young patients [16]. The patient in our second case showed severe loss of enamel in permanent dentition with excess space in the upper and lower jaws. Therefore, indirect composite crowns were used to rebuild the vertical dimension. Previous impressions and cooperation with the dental technician made rehabilitation of the chewing system easier. The interdental spaces could be used to avoid grinding of the teeth. The process of preparing the teeth could protect the hard dental tissue. Other authors have described direct composite restoration in combination with a wax-up to restore complex cases [17]. However, this treatment is very time-consuming and demanding. In the case of adhesive restorations, it should be noted that normal conditioning (etching pattern, effectiveness of the adhesive system) is not possible in affected enamel, so very early failure and loss of restorations are often recorded [18]. In addition, the remaining enamel can repeatedly flake off, so the corresponding tooth will need a new restoration or the existing one will need to be expanded [4].

Generally, a successful therapy concept is based on a close recall program with oral hygiene instruction, remotivation, and fluoride application. In this way, carious lesions or restorative defects and gingivitis can be prevented. A limitation of our study is that up to now there has only been a follow-up of 6 months after intervention. However, since the parents’ compliance with the follow-up appointments is considered reliable, we expect a good prognosis with regard to the aspects mentioned above. In addition, the use of electric toothbrushes should also be considered regarding the practical implementation of home oral hygiene. A study by Preda et al. showed that electric toothbrushes were superior to manual toothbrushes in plaque removal. Therefore, rotating–oscillating or sonic-action heads should be recommended for patients with difficult hygienic conditions to avoid bacterial infiltration [19].

Due to the extensive treatment needs, complex treatment measures, and often age-related insufficient cooperation, comprehensive rehabilitation under general anesthesia cannot be avoided. Overall, the patients report less sensitivity, better oral hygiene ability, and better quality of life.

## 4. Conclusions

The cases described in this paper show the complexity of the dental care of structural anomalies of genetic origin. Patients with hereditary structural anomalies require close lifelong dental care to maintain the therapeutic results.

## Figures and Tables

**Figure 1 ijerph-18-07204-f001:**
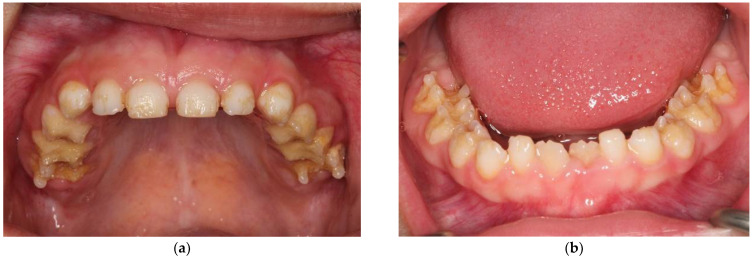
A 5 ½-year-old patient with AI type I: (**a**) upper teeth; (**b**) lower teeth. Clinical exanimation revealed pain and hypersensitivity in yellow teeth and loss of dental structure (pits).

**Figure 2 ijerph-18-07204-f002:**
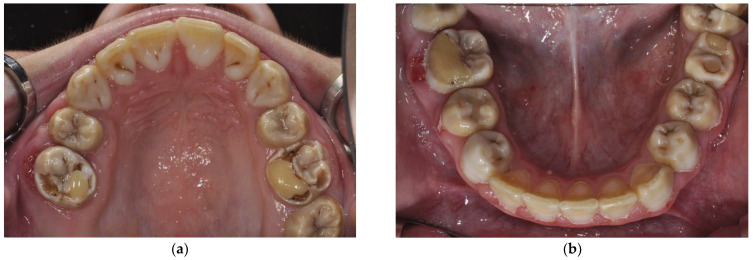
A 14-year-old patient with hypomaturation of AI: (**a**) upper teeth; (**b**) lower teeth. Clinical exanimation revealed yellow teeth affecting oral health related quality of life.

**Figure 3 ijerph-18-07204-f003:**
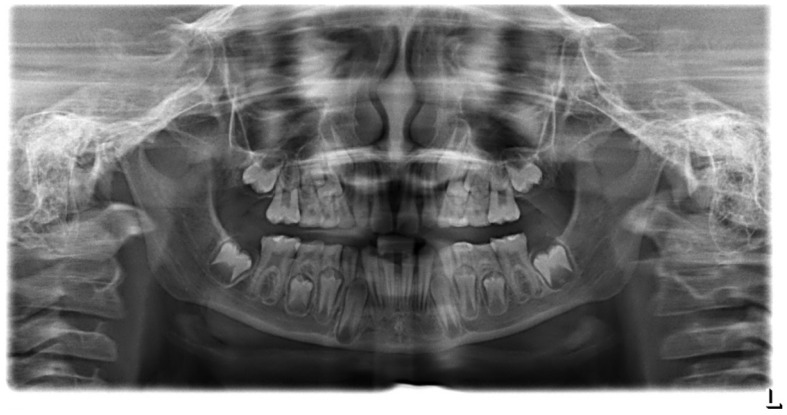
Panoramic X-ray of 7-year-old patient.

**Figure 4 ijerph-18-07204-f004:**
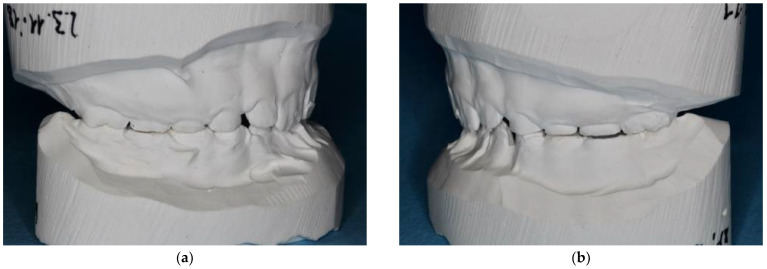
Plaster models of upper and lower jaws showing loss of tooth wear because of attrition and abrasion (physical tooth wear): (**a**) right side; (**b**) left side.

**Figure 5 ijerph-18-07204-f005:**
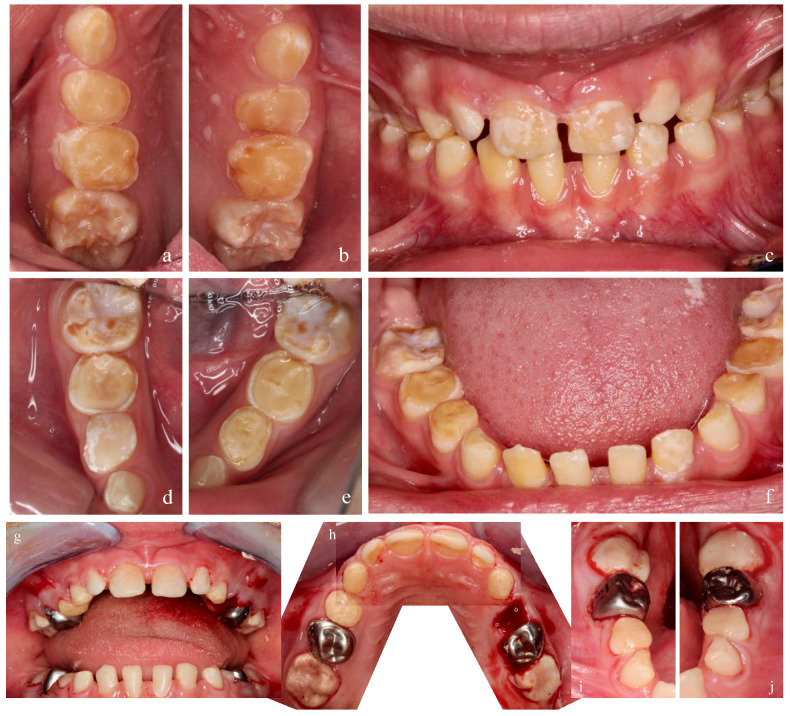
A 7-year-old patient with AI type I. (**a**–**f**) Preoperative situation with multiple substance defects on all teeth; (**c**) after fluoride varnish application. (**g**–**j**) Postoperative result. Oral surgery was performed under general anesthesia, with stainless steel crowns applied to second primary molars and adhesive filling materials in first primary molars, first molars and anterior teeth.

**Figure 6 ijerph-18-07204-f006:**
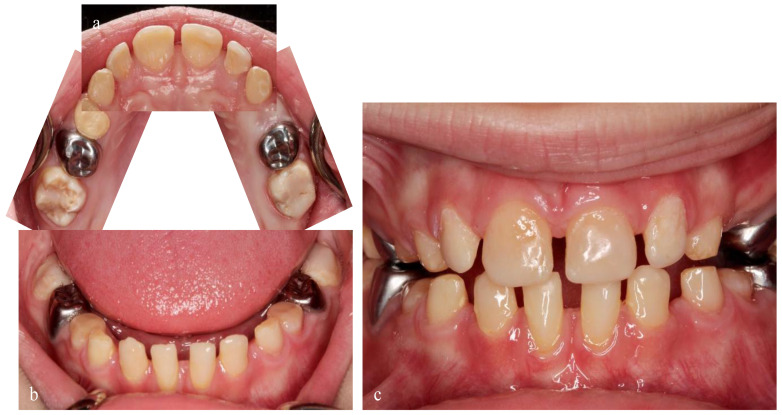
Follow-up 6 months after restoration under general anesthesia: (**a**) upper teeth; (**b**) lower teeth; (**c**) front teeth. All restorations in situ, no abnormalities.

**Figure 7 ijerph-18-07204-f007:**
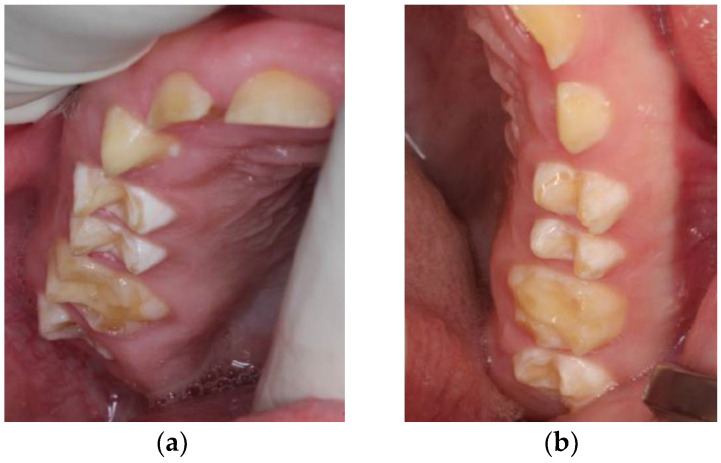
A 12-year-old patient showing clinical signs of AI type II: (**a**,**b**); upper teeth (**c**) front teeth. Besides the yellow color, gaps between teeth are predominant.

**Figure 8 ijerph-18-07204-f008:**
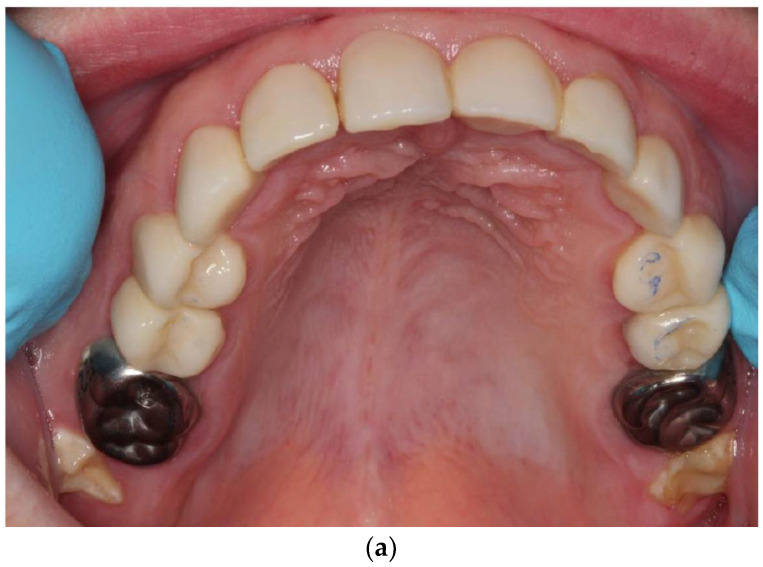
A 12-year-old patient with AI type II: (**a**) upper teeth; (**b**) lower teeth. Oral surgery was performed under general anesthesia using indirect composite restorations and stainless steel crowns.

## Data Availability

Not applicable.

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
