# Peer review of "Management of Amelogenesis Imperfecta in Childhood: Two Case Reports"

_ijerph, 2021, doi:10.3390/ijerph18137204_

Round 1

Reviewer 1 Report

This manuscript aimed to report two cases of treatment for the amelogenesis imperfecta patients. The authors described the treatments step-by-step. In case 2, the occlusal vertical dimension is extremely lower than unaffected ones. How did the authors/clinicians decide the his appropriate dimension? After the treatment and during the adaption period, did this patient have any symptom in the temporomandibular joint?

Author Response

Response letter:

Submission of revised manuscript: ijerph-1250767

Title: Management of Amelogenesis imperfecta in childhood: two case reports

Dear editor,

thank you for provision of the reviewers’ comments which were very helpful and well received. According to their suggestions, the following changes have been made.

Thank you very much for your kind consideration for this manuscript.

Best regards.

Reply to Reviewer 1:

Thank you for your comments. We have made following changes:

  1. In case 2, the occlusal vertical dimension is extremely lower than unaffected ones. How did the authors/clinicians decide this appropriate dimension?

Our response: Thank you for your comment. We totally agree. A jaw relation determination could not be performed due to the patient's insufficient compliance.  A so-called “hand bite” was also not possible.

Therefore, the assessment of the unconscious spacing between the upper and lower jaw with relaxed jaw muscles (habitual lower jaw position) was presented via the neuromuscular method. The indirect restorations were then fabricated on plaster models after articulation according to average values. We added these information in the text presenting case 2.

See revised text, page 6, line 145-149

  1. After the treatment and during the adaption period, did this patient have any symptom in the temporomandibular joint?

Our response: Thank you for your comment. Adaptation to the new mandibular posture after insertion worked without problems. We detected no symptoms in the temporomandibular region after the 3 and 6 months recall. We added this information to case report 2.

Revised text: See text, page 6, line 159-160

Reviewer 2 Report

The manuscript presents a complete structure, with introduction, materials and methods, results and conclusions. The cases of amelogenesis given the genetic components begin to be more and more widespread, therefore I would like to point out some aspects to be reviewed in the manuscript:
- Citing general anesthesia, did you have parents sign the informed consent? possibly attach it
- mention all the home oral hygiene aids to reduce the risk of bacterial infiltration, I would like to point out a reference:

The efficacy of powered oscillating heads vs. Powered sonic action heads toothbrushes to maintain periodontal and peri-implant health: A narrative review

International Journal of Environmental Research and Public Health, 2021, 18(4), pp. 1–17, 1468

- add in addition to fluorine other remineralizing systems such as acp and hydroxyapatite
- being that they are pedodontic patients, insert evidence on the use of probiotics to reduce the incidence of dysbiosis that induces lesions in hard and soft tissues
-improve the quality of the photos, especially in figure 5 E

Author Response

Response letter:

Submission of revised manuscript: ijerph-1250767

Title: Management of Amelogenesis imperfecta in childhood: two case reports

Dear editor,

thank you for provision of the reviewers’ comments which were very helpful and well received. According to their suggestions, the following changes have been made.

Thank you very much for your kind consideration for this manuscript.

Best regards.

Reply to Reviewer 2:

Thank you for your comments. We have made following changes:

  1. Citing general anesthesia, did you have parents sign the informed consent? possibly attach it

Our response: Thank you for your comment. We attached informed consent.

  1. mention all the home oral hygiene aids to reduce the risk of bacterial infiltration, I would like to point out a reference:

The efficacy of powered oscillating heads vs. powered sonic action heads toothbrushes to maintain periodontal and peri-implant health: A narrative review

Our response: Thank you for your comment. We integrated this important aspect in the discussion section and added the literature.

Revised text: See text, page 8, line 331-336

  1. add in addition to fluorine other remineralizing systems such as acp and hydroxyapatite

Our response: Thank you for your comment. Except for fluoride, neither casein phosphopeptide amorphous calcium phosphate (CPP-ACP), tricalcium phosphate or self-assembling peptides were used as remineralising agents.

  1. being that they are pedodontic patients, insert evidence on the use of probiotics to reduce the incidence of dysbiosis that induces lesions in hard and soft tissues

Our response: Thank you for your comment. Probiotics can have both preventive and therapeutic effects. In the context of oral health, probiotic preparations can prevent the oral biofilm from being "stressed" by the environment and maintain a stable symbiosis associated with health. However, beneficial bacteria can also "repair" a dysbiotic biofilm associated with disease.  Although we are aware of this information, the use of probiotics is not in line with our oral health recommendations.

  1. improve the quality of the photos, especially in figure 5 E

Our response: Thank you for your comment. We attached the original photos and reorganised the figure 5 for better quality.

Reviewer 3 Report

paper has presentation issues, check for gramar and style issues abstract, add the age of patients and follow up period introduction, make it more interestin add a table for 4 types of AI add page number, line number change the ''direct filling therapies ''to ''composite veneer'' case 1, do you have longer followup? Figure 3. X-ray of the same patient. correct to 'Figure 3. panoramic X-ray of the same patient.' Figure 4. a and b: Plaster models of the upper and lower jaws showing vertical loss. what do you mean, ''tooth wear''? page 4, right at the end'The gap regio 64 was stable, so that a gap holder was not fabricated (Figure 6a-c' revise??? Figure 7. a-c: 12-year-old patient showing clinical signs of AI hypomaturation. Besides the yellow color, the gaps between the teeth are predominant. revise 'AI hypomaturation'???? can you provide a better occlusal view talk more about compomer (Dyract) and explain the new GIC materials and new methods used to enhance the physical properties of GIC.Compomers have poorer mechanical properties than dental composites, with a lower compressive, flexural and tensile strength. Therefore, compomers are not an ideal material for load bearing restorations

Author Response

Response letter:

Submission of revised manuscript: ijerph-1250767

Title: Management of Amelogenesis imperfecta in childhood: two case reports

Dear editor,

thank you for provision of the reviewers’ comments which were very helpful and well received. According to their suggestions, the following changes have been made.

Thank you very much for your kind consideration for this manuscript.

Best regards.

Reply to Reviewer 3:

Thank you for your comments. We have made following changes:

  1. check for grammar and style issues abstract

Our response: Thank you for your comment. We had checked the article by a native speaker and  checked the manuscript again for English improvement. If there are unclear formulation to be revised, please point out specific possibilities for improvement and we will gladly correct them.

  1. add the age of patients and follow up period introduction

Our response: Thank you for your comment. We added this information to the introduction section.

See revised text, page 2, line 70-71

  1. add a table for 4 types of AI

Our response: We have based our differentiation on the clinical appearance of the four forms of amelogenesis imperfecta and have not considered a classification according to heredity and the resulting biochemical structural changes. Therefore, we have dispensed with the tabular discussion.

  1. change the ''direct filling therapies ''to ''composite veneer'' case 1

Our response:

Thank you for your comment. The teeth were reconstructed by means of filling therapy without the use of shaping aids. In contrast to case 2, only direct restorations were performed.

  1. Do you have longer followup?

Our response: Thank you for your comment. The patients are both in recall at our polyclinic. The described follow-ups were the last appointments.

  1. Figure 3. X-ray of the same patient. correct to 'Figure 3. panoramic X-ray of the same patient.'

Our response: Thank you for your comment. We totally agree and corrected the legend.

See revised text, page 3, line 86

  1. Figure 4. a and b: Plaster models of the upper and lower jaws showing vertical loss. What do you mean, ''tooth wear''?

Our response: Thank you for your comment. This might be misleading. The plaster models of the upper and lower jaw showing vertical loss because physical tooth wear meaning abrasion and attrition. We added these terms in the legend.

See revised text, page 4, line 88-89

  1. Page 4, right at the end 'The gap regio 64 was stable, so that a gap holder was not fabricated

Our response: Thank you for your comment. We apologize, that the sentence was misleading. In accordance with the DGZMK guideline (Indication for the fabrication of space maintainers after early deciduous tooth loss) we measured the gap. Since there was no space narrowing, we did not need to fabricate a space maintainer. We reorganized this aspect in the text.

See revised text, page 4, line 107-108

  1. Figure 6a-c' revise???

Our response: Thank you for your comment. We added the original photos for better quality.

  1. Figure 7. a-c: 12-year-old patient showing clinical signs of AI hypomaturation. Besides the yellow color, the gaps between the teeth are predominant. revise 'AI hypomaturation'????

Our response: Thank you for your comment. We totally agree - to keep it consistent, we changed the term to AI type II.

See revised text, page 5, line 135

  1. Can you provide a better occlusal view talk more about compomer (Dyract) and explain the new GIC materials and new methods used to enhance the physical properties of GIC. Compomers have poorer mechanical properties than dental composites, with a lower compressive, flexural and tensile strength. Therefore, compomers are not an ideal material for load bearing restorations

Our response: Thank you for your comment. We totally agree. We reinforced and discussed these important aspect sin the text.

See revised text- discussion section, page 7, line 197-205

Round 2

Reviewer 3 Report

the major issue is the grammar and style/presentation of the paper, you need to get a native read and revise it, starting from abstract

the follow-up of 3-6 months is very short! discuss this in the paper

more information is needed for case 1 on the state of u/l 2-2 and u/l 6s, how bad they were affected, the is spacing as well, mention it, what orthodontic intervention may be needed in future as u/l canines seem to be short of space?

figure 5, explain at what follow-up you took this images, would be great if you add longer follow, other OPGs taken, later on, some comments on the structure of u/l 345s,

figure 5c does not show much loss of vertical dimension, just mention tooth wear, better images showing u/l 6s are needs add if you have any

second case report needs an OPG

Author Response

Response letter:

Submission of revised manuscript: ijerph-1250767

Title: Management of Amelogenesis imperfecta in childhood: two case reports

Dear editor,

thank you for provision of the reviewers’ comments which were very helpful and well received. According to their suggestions, the following changes have been made.

Thank you very much for your kind consideration for this manuscript.

Best regards.

Reply to Reviewer 3:

Thank you for your comments. We have made following changes:

  • The major issue is the grammar and style/presentation of the paper, you need to get a native read and revise it, starting from abstract

Our response: Thank you for your comment. We already checked our manuscript by an native speaker but decided to undergo extensive English revision proposed by the MDPI editing service (certificate enclosed).

  • The follow-up of 3-6 months is very short! Discuss this in the paper

Our response: We totally agree. For both patients, the last appointment was the 6-month recall. However, both are in the further follow-up programme with the next planned presentation after another 3 months (9 months after ITN). We discussed this aspect in the discussion section.

See revised text, page 8, line 241-249.

  • More information is needed for case 1 on the state of u/l 2-2 and u/l 6s, how bad they were affected, the is spacing as well, mention it, what orthodontic intervention may be needed in future as u/l canines seem to be short of space?

Our response: We added more information on the affected upper and lower front and first permanent molars.

See revised text, page 3, line 80-82.

Regarding orthodontic intervention, we discussed this aspect radiographically in an interdisciplinary consultation deciding that up to today there is no space problem. If there is a lack of space after the primary teeth have been changed, an interdisciplinary consultation is possible at any time.

  • Figure 5, explain at what follow-up you took this images, would be great if you add longer follow, other OPGs taken, later on, some comments on the structure of u/l 345s,

Our response: Thank you for your comment. Images were directly taken after operation (intra/ postoperative). You can see 6 months follow-up at figure 6. There are no more OPGs because we need justifying indication.

  • Figure 5c does not show much loss of vertical dimension, just mention tooth wear, better images showing u/l 6s are needs add if you have any

Our response: Thank you for your comment. We changed the termini in tooth wear.

See revised text, page 3, line 84, 88-89.

Furthermore, we expanded figure 5 adding two images showing 36 and 46.

See revised text, page 4, line 107-109.

  • second case report needs an OPG

 Our response: Thank you for your comment. We totally agree, but because of insufficient compliance of patient and inability to take X-rays under general anesthesia, we have no panoramic X-ray.